# Detection and spread of high pathogenicity avian influenza virus H5N1 in the Antarctic Region

Ashley C. Banyard [1,2] ✉, Ashley Bennison[3], Alexander M. P. Byrne[1,4], Scott M. Reid[1], Joshua G. Lynton-Jenkins[1,2], Benjamin Mollett[1], Dilhani De Silva[1], Jacob Peers-Dent [1], Kim Finlayson[5], Rosamund Hall[3], Freya Blockley[3], Marcia Blyth[3], Marco Falchieri[1], Zoe Fowler[6], Elaine M. Fitzcharles[3], Ian H. Brown[1,2] & Joe James [1,2] ✉

Until recent events, the Antarctic was the only major geographical region in which high pathogenicity avian influenza virus (HPAIV) had never previously been detected. Here we report on the detection of clade 2.3.4.4b H5N1 HPAIV in the Antarctic and sub-Antarctic regions of South Georgia and the Falkland Islands, respectively. We initially detected H5N1 HPAIV in samples collected from brown skuas at Bird Island, South Georgia on 8th October 2023. Since this detection, mortalities were observed in several avian and mammalian species at multiple sites across South Georgia. Subsequent testing confirmed H5N1 HPAIV across several sampling locations in multiple avian species and two seal species. Simultaneously, we also confirmed H5N1 HPAIV in southern fulmar and black-browed albatross in the Falkland Islands. Genetic assessment of the virus indicates spread from South America, likely through movement of migratory birds. Critically, genetic assessment of sequences from mammalian species demonstrates no increased risk to human populations above that observed in other instances of mammalian infections globally. Here we describe the detection, species impact and genetic composition of the virus and propose both introductory routes and potential long-term impact on avian and mammalian species across the Antarctic region. We also speculate on the threat to specific populations following recent reports in the area.

Following the emergence and global expansion of A/goose/Guangdong/1/96 (GsGd)-lineage H5 high pathogenicity avian influenza viruses (HPAIV) there have been repeat epizootics in wild birds and poultry populations globally. This lineage spread globally, evolving into diverse clades classified according to the H5 haemagglutinin (HA) gene phylogeny[1]. In the autumn of 2021, the situation escalated considerably with the detection of a clade 2.3.4.4b HPAIV subtype H5N1 in Europe[2]. Subsequently, two unprecedented epizootic waves with this lineage in 2021/22 and 2022/23 were associated with mass mortality events in wild birds together with unprecedented numbers of incursions into poultry premises[2–6]. High levels of viral adaptation to wild bird species[7], and increased fitness advantage through continued

[1]Department of Virology, Animal and Plant Health Agency (APHA-Weybridge), Woodham Lane, Addlestone, Surrey KT15 3NB, UK. [2]WOAH/FAO International Reference Laboratory for Avian Influenza, Animal and Plant Health Agency (APHA-Weybridge), Woodham Lane, Addlestone, Surrey KT15 3NB, UK. [3]British Antarctic Survey, Madingley Road, Cambridge CB3 0ET, UK. [4]Worldwide Influenza Centre, The Francis Crick Institute, Midland Road, London NW1 1AT, UK. [5]KEMH Pathology and Food, Water & Environmental Laboratory, St Mary's Walk, Stanley FIQQ 1ZZ, Falkland Islands. [6]Department of Agriculture, Bypass Road, Stanley FIQQ 1ZZ, Falkland Islands. ✉e-mail: ashley.banyard@apha.gov.uk; joe.james@apha.gov.uk

genetic reassortment[8] likely underpin the broad impact infection has had across many avian species[3]. This wide host range has facilitated the transmission of the lineage across a large geographic area, including from Europe to North America[9,10], where it has since rapidly expanded its range southward into South America via migratory avian species. Incursion into South American countries, starting in November of 2022, represented the first recorded instances of GsGd-lineage H5 HPAIV in the region[11–13]. Mass mortality events in the region have been particularly severe and reported across several different bird species in addition to marine mammals[3,11,14], highlighting the extensive ecological impact of HPAIV and the ongoing threat it presents to naïve hosts.

The Antarctic region includes the ice shelves, waters, and all the island territories in the Southern Ocean situated inside of the 'Antarctic Convergence' or 'Antarctic Polar Front', a marine belt encircling Antarctica where Antarctic waters meet those of the warmer sub-Antarctic[15]. Several islands are located inside the Antarctic region, including South Georgia, while the Falkland Islands among others are located outside the Antarctic Convergence in the sub-Antarctic zone (Fig. 1). There have been no previous reports of HPAIV inside the Antarctic region[16,17]. Antarctica and the sub-Antarctic islands possess unique ecosystems which support the population strongholds of several avian and marine mammal species. The relative isolation of these islands from human populations has provided species across the Antarctic with only limited protection from anthropogenic environmental change[18]. Indeed, wildlife populations in the Antarctic face a broad range of challenges from introduced species[19], to longline fisheries[20,21], and rapid climate change[22–24]. Several native bird species including wandering albatross (*Diomedea exulans*), macaroni penguins (*Eudyptes chrysolophus*), grey-headed albatross (*Thalassarche chrysostoma*), and white-chinned petrel (*Procellaria aequinoctialis*), are listed as either vulnerable or endangered[25]. Iconic long-lived species with late maturity, such as albatross, exhibit low resilience to rapid increases in population mortality[26]. High mortality disease outbreaks therefore represent a substantial threat to already vulnerable seabird populations[27,28].

While geographically isolated, several Antarctic seabird species routinely range between the South Atlantic and Southern Ocean, visiting the South American coast to either forage or overwinter[29]. Brown skuas (*Stercorarius antarcticus*), kelp gulls (*Larus dominicanus*), southern giant petrel (*Macronectes giganteus*), and snowy sheathbills (*Chionis albus*) have previously been identified as potential vectors of infectious pathogens into this vulnerable ecosystem due to their migratory traits, scavenging behaviour, and previously identified roles as carriers of low pathogenicity avian influenza viruses (LPAIV)[30–37].

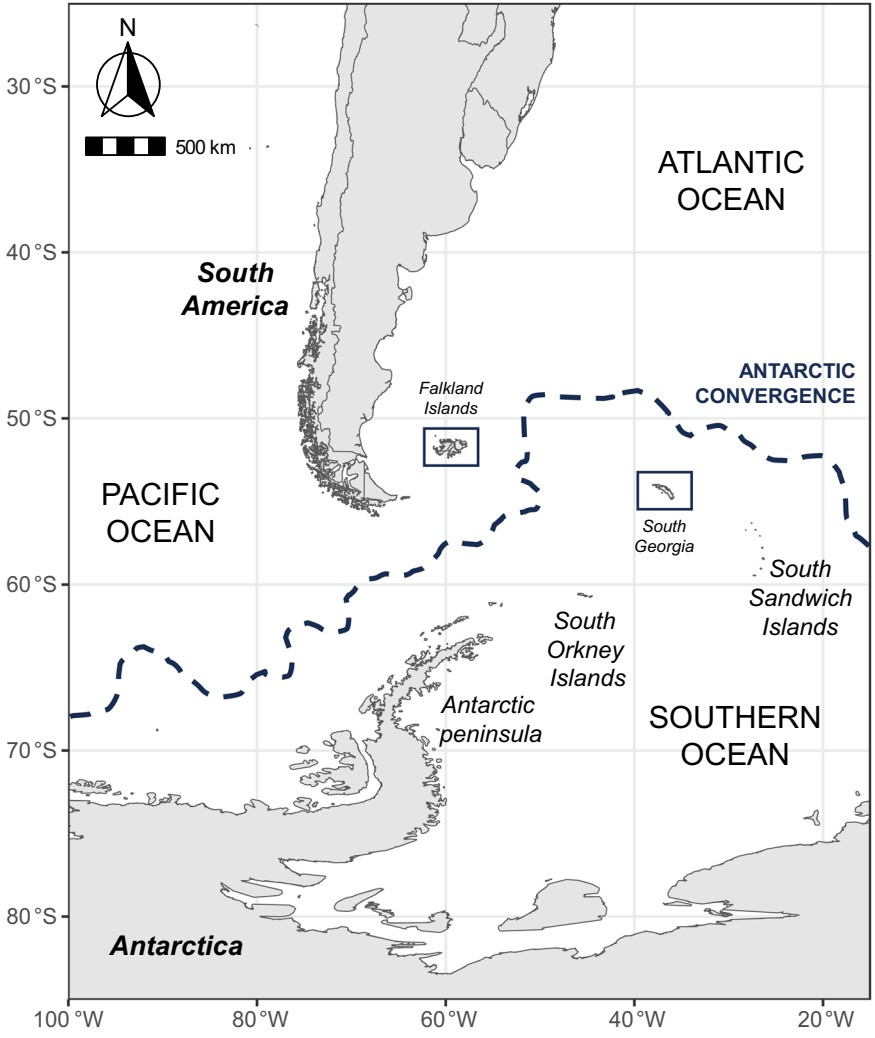

**Fig. 1 | The Antarctic region showing locations of South Georgia and the Falkland islands.** Map showing the locations of South America, the Falkland Islands, South Georgia, the Antarctic peninsula, and other major islands in the Antarctic region. The location of the Antarctic Convergence (Antarctic Polar Front) is indicated by a blue hashed line.

Evidence of LPAIV in the Antarctic region was first identified from serological studies in the 1980s[38–40]. A range of subtypes have since been reported (H1[40], H3[31], H4[35], H5[41], H6[32], H7[39], H9[31], and H11[33]) including genetic analysis of the influenza virus subtypes H4N7[35], H5N5[33,41], H6N8[32] and H11N2[33,42,43]. In contrast to the more prevalent H11N2 influenza viruses, which likely circulate silently through local populations, H4-H6 subtypes were found to share high sequence similarity with viruses from South America, indicating more recent introduction events[32,33,35]. Evidence of LPAIV transmission to the continent from the Americas demonstrates the elevated risk of clade 2.3.4.4b H5 HPAIV being introduced to the Antarctic, encouraging researchers in 2022 to employ additional biosecurity measures while maintaining surveillance activities[30,44]. During the austral summer of 2022/23, sampling and surveillance was conducted at several sites in the Antarctic region, and as of March 2023, HPAIV had not been detected[16].

Here we describe the observation of morbidity and mortality events across different species as well as the positive detection of H5N1 HPAIV in a variety of species in South Georgia, inside the Antarctic region and the sub-Antarctic Falkland Islands. We detail the suspicion, diagnostic evaluation, and clinical presentations of HPAIV in the region. Genetic analysis is used to characterise potential introduction routes and the consequences of HPAIV circulation in this region are considered.

## Results
### Case description and virus detection
Provision for diagnostic investigation of avian influenza is limited in the Antarctic and sub-Antarctic regions. For some samples in the description below, local RT-PCR testing at the KEMH Pathology and Food, Water & Environmental Laboratory on the Falkland Islands was used to screen for the presence of avian influenza virus H5N1 viral RNA (vRNA). This was initially undertaken for a Southern Fulmar as described below and this detection, alongside the increase clinical disease and mortalities observed on South Georgia triggered shipment of samples to the World Organisation for Animal Health (WOAH) and Food and Agriculture Organisation (FAO) International Reference Laboratory for avian influenza, swine influenza and Newcastle disease virus at the Animal and Plant Health Agency (APHA), Weybridge, UK for confirmatory and further diagnostic evaluation. An overview of both observations and testing is detailed below.

On the 17th September, researchers of the British Antarctic Survey (BAS) on Bird Island, South Georgia (an island approximately 0.5 km Northwest of the main island of South Georgia) (Fig. 2A) discovered a single southern giant petrel showing behaviours indicative of loss of coordination, neurological twitching, and lethargy. This individual was observed being preyed and scavenged upon by brown skuas and other southern giant petrels (*Macronectes giganteus*). On 8th October, brown skuas were observed in the same locality showing lethargy, neck spasms, twitching, and an inability to fly, and by 10th October, bird mortality was seen on Bird Island, with the highest number of mortalities occurring at the roosting site of non-breeding birds. Swab samples were collected from the three brown skua (*Stercorarius antarcticus*) on 8th October 2023 and a further brown skua on the 11th October which were all later found dead on Bird Island. These initial sampling events all yielded PCR positivity for HPAIV H5N1. The series of sampling events and species testing positive from this point onwards are detailed in Table 1 and Supplementary Table S1.

Briefly, both brown skuas and kelp gulls then tested positive at two different locations on South Georgia (Hound Bay and St. Andrews Bay) on the 30th October with further gulls and skuas (Moltke Harbour) swabbed a day later (31st October) also testing positive. Further positive samples were taken on the 3rd (Harpon Bay) and 8th (Penguin River) November from both brown skuas and kelp gulls. Further escalation in mortality occurred by 17th November 2023, when 57

skuas were observed to have died at Bird Island although samples could not be retrieved. On the 27th of November a South Georgia shag tested positive from King Edward Cove alongside an Antarctic tern sampled dead on the 6th of December.

Alongside avian species, in early December clinical disease consistent with mammalian infection with HPAIV was observed in colonies of southern elephant seal (*Mirounga leonine*) and Antarctic fur seal (*Arctocephalus gazella*) at Jason Harbour on South Georgia (Supplementary Table S2). Clinical presentation included difficulty breathing, with coughing and short sharp breath intake. Individuals also showed accumulation of viscous fluid around the nasal passage. Five southern elephant seals and 1 Antarctic fur seal from Jason Harbour collected on the 9th December tested positive for the virus (Supplementary Table S2).

Overall, between the initial detection of HPAIV on the 8th of October and 9th December, from South Georgia a total of 33 avian carcasses and 17 mammalian carcasses were sampled including representatives of five different avian species and two mammalian species across eight different locations. Of these, 66% ($n = 28/33$ avian carcasses; $n = 6/17$ mammalian carcasses) tested positive for HPAIV H5N1.

Concurrent to the events occurring on South Georgia, on 30th October, a southern fulmar (*Fulmarus glacialoides*) was reported dead on Stanley, Falkland Islands and tested positive for the virus (Fig. 2B). Numerous other swabs were taken from different avian species across the Falkland Islands (Supplementary Table 1) but only three black-browed albatross (*Thalassarche melanophris*) tested positive from Saunders Island ($n = 1$) and Steeple Jason ($n = 2$) being collected on the 19th and the 26th of November, respectively. In total between the 30th October and 10th December samples were taken and tested from a total of 13 carcasses from nine different bird species with 31% testing positive for HPAIV H5N1 (Table 1 and Supplementary Table S1).

### Virus isolation
Wherever samples tested positive for HPAIV H5N1, attempts to isolate viable virus were undertaken. From the samples collected from South Georgia, virus isolation was successful from two brown skuas and a kelp gull taken from Hound Bay on the 30th October, a brown skua and kelp gull collected from Moltke harbour on the 31st October, a kelp gull from Harpon Bay on the 3rd November and both a South Georgia shag and an Antarctic tern from King Edward Cove on the 27th November and the 6th December, respectively.

Virus isolation was also successful from swabs taken from carcasses found dead on the Falklands Islands, including from samples from two southern fulmar collected on the 30th October and the 13rd November from the Falklands Islands, and from black-browed albatross from Saunders Island and Steeple Jason on the 19th November and the 26th November, respectively.

From a mammalian perspective, no viable virus could be recovered from any of the samples testing positive by the generic AIV, HPAIV H5-specific assay and N1-specific rRT-PCR assays[45–48] (Supplementary Table 2).

### Genomic and phylogenetic analysis
Whilst successful virus isolation gave material for full virus genomic sequencing and analysis, wherever possible, samples for which only rRT-PCR detection was possible were also assessed for full genome generation and sequence analysis. A total of 20 full genomes were generated from avian samples and a further three full genomes were generated from samples collected from seals. These included ten full virus genomes from brown skuas (from Bird Island, Hound Bay, Moltke Harbour, and Penguin River); six full virus genomes from kelp gulls (from Hound Bay, Moltke Harbour, Harpon Bay, and Penguin River); and both a South Georgia shag and an Antarctic tern (both from King Edward Cove). A further avian full virus genome was generated from

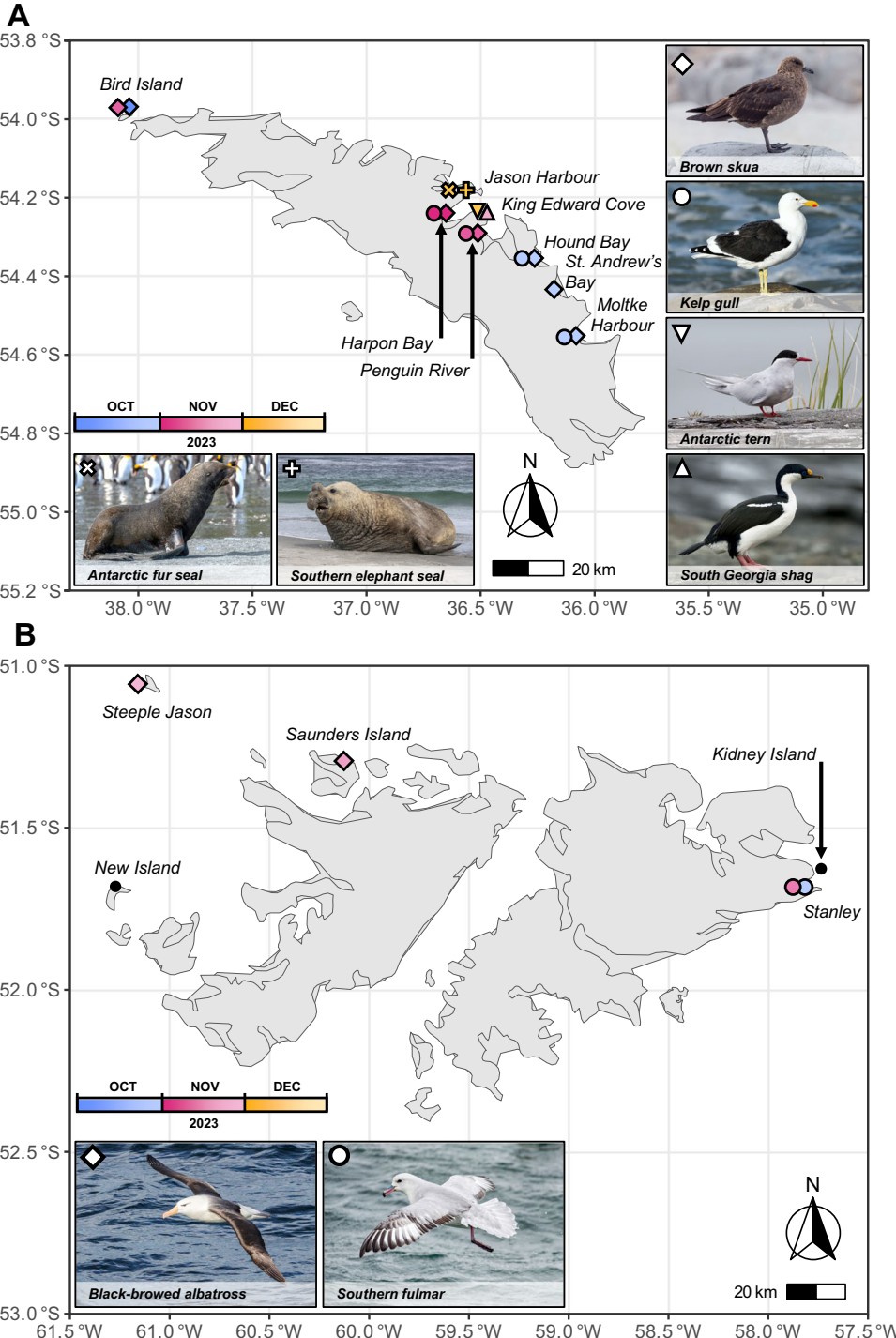

**Fig. 2 | Geographical distribution of H5N1 HPAIV detections from South Georgia and The Falkland Islands. A** Map showing the location, species, and date of influenza positive samples collected from South Georgia. **B** Map showing the location, species, and date of influenza positive samples collected from the Falkland Islands. Locations with mortalities but no positive samples are also labelled.

two Southern Fulmars, both from Stanley Island on the Falklands Islands (Supplementary Table S1).

From a mammalian perspective we generated two full virus genomes from southern elephant seals (both from Jason Harbour); and one full virus genome from an Antarctic fur seal (from Jason Harbour) (Supplementary Table S2).

Comparison of the full data set generated from samples described revealed that they shared 99.86–100% nucleotide identity across all eight influenza gene segments (Supplementary Table S3). The

sequences generated were then analysed with representative global H5N1 clade 2.3.4.4b full-genome sequences to assess genetic ancestry by inferring maximum-likelihood phylogenetic trees (Supplementary Fig 1). The sequences from South Georgia and the Falkland Islands clustered with H5N1 viral sequences obtained from South America between October 2022 and October 2023, in particular Argentina, Uruguay, Peru, and Chile across all gene segments. All sequences from South Georgia and the Falkland Islands were genotyped according to the United States of America H5N1 schema, given the spread of these

**Table 1 | Summary of H5 HPAIV testing results from avian and mammalian species from South Georgia and the Falkland Islands**

| Location | Common name | Scientific name | H5 HPAIV +ve animals (total tested) | Sequences obtained |
|---|---|---|---|---|
| South Georgia | Brown skua | *Stercorarius antarcticus* | 15 (15) | 9 |
| | Kelp gull | *Larus dominicanus* | 11 (11) | 6 |
| | Antarctic tern | *Sterna vittata* | 1 (1) | 1 |
| | South Georgia shag | *Leucocarbo georgianus* | 1 (5) | 1 |
| | King penguin | *Aptenodytes patagonicus* | 0 (1) | 0 |
| | Southern elephant seal | *Mirounga leonina* | 5 (13) | 2 |
| | Antarctic Fur Seal | *Arctocephalus gazella* | 1 (4) | 1 |
| Falkland islands | Black browed albatross | *Thalassarche melanophris* | 2 (4) | 0 |
| | Southern fulmar | *Fulmarus glacialoides* | 2 (2) | 2 |
| | Austral thrush | *Turdus falcklandii* | 0 (1) | 0 |
| | Chicken | *Gallus gallus domesticus* | 0 (1) | 0 |
| | Falkland steamer duck | *Tachyeres brachypterus* | 0 (1) | 0 |
| | Grey-backed storm petrel | *Garrodia nereis* | 0 (1) | 0 |
| | Imperial shag | *Leucocarbo atriceps* | 0 (1) | 0 |
| | Rock shag | *phalacrocorax magellanicus* | 0 (1) | 0 |
| | Southern rockhopper penguin | *Eudyptes chrysocome* | 0 (1) | 0 |

viruses from North to South America in early 2022[11], and found to be part of the B3.2 genotype[49].

To further investigate the introduction of H5N1 HPAIV into South Georgia and the Falkland Islands, representative H5N1 clade 2.3.4.4b HA sequences from North and South America were used to perform time-resolved phylogenetic analysis (Fig. 3A and Supplementary Fig 2). This analysis provided evidence for distinct, separate introductions of H5N1 into South Georgia and the Falkland Islands, with both sets of sequences sharing a common ancestor with sequences from South America dating back to July 2023 (Falkland Islands, range: December 2022 to November 2023) and August 2023 (South Georgia, range: February November 2023). However, both sets of sequences produced long branch lengths compared to South American sequences, potentially due to unsampled ancestry. The sequences from South Georgia formed a single monophyletic group suggestive of a single introduction, whilst those from the Falkland Islands formed a paraphyletic group, with sequences from Argentina and Brazil. To investigate the source of these introductions, a Bayesian stochastic search variable selection (BSSVS) analysis was performed using the country or location from which the sequences originated as a discrete trait and then the support for these transmissions was quantified using Bayes factors (BFs). This analysis demonstrated that Argentina was the likely source of H5N1 HPAIV for both South Georgia (moderate support with a BF of 3–10) and the Falkland Islands (very strong support with a BF of 30–100) (Fig. 3B). The spread of H5N1 HPAIV within South Georgia was then also investigated using the same approach (Fig. 3C). This analysis further supported that the initial introduction into South Georgia occurred on Bird Island, before onward spread to Penguin River (definitive support with a BF > 100), Hound Bay (strong support with a BF of 10–30) and Moltke Harbour (moderate support with a BF of 3–10). Interestingly, both Harpon Bay and Jason Harbour did not demonstrate a statistically relevant connection to Bird Island based on the BSSVS analysis, but also there was no connection between these sampling locations and any outside of South Georgia either. Nevertheless, this demonstrates that after introduction into South Georgia, H5N1 HPAIV was then spread onward to multiple locations across the islands, whereby it infected multiple avian and mammalian species.

Alongside interrogation of phylogenetic ancestry, the sequences derived from all species were assessed for the presence or absence of adaptive mutations that may indicate adaptation to replication in mammals. From the mammalian samples, only a single amino acid change of interest (PB2 E627K in one southern elephant seal sequence

from Jason Harbour) was observed. None of the sequences contained any of the other PB2 mutations of particular interest (i.e., T271A, K526R, Q591K or D701N) or any other mutations associated with adaptation to replication in mammals. Interestingly, a kelp gull sequence from Harpon Bay displayed a single amino acid change of interest, PB2 D701N, but no others. Both southern fulmar sequences obtained from the Falkland Islands contained Q591K and D701N mutations. There were also no mutations in either mammalian or avian species that would affect the susceptibility of this virus to antivirals (Supplementary Table S4). WGS read coverage for all sequences generated can be found in Supplementary Table S5.

## Discussion

Since the emergence and global expansion of Gs/Gd-lineage H5Nx HPAIV in 1996, Antarctica and Oceania were the only two continents in which it had not been detected. Moreover, until now, the Antarctic region remained the only major geographical region in which HPAIV had never been detected. The island of South Georgia lies in the Southern Ocean inside the Antarctic convergence, a marine belt encircling Antarctica which defines the Antarctic Region. The island is an area of high biodiversity and high conservation priority with multiple species being defined as vulnerable to the incursion of infectious diseases[50–52]. The Falkland Islands constitute a remote cluster of islands in the South Atlantic Ocean situated approximately 1500 km to the west of South Georgia. The Falkland Islands are situated outside of the Antarctic convergence, in the sub-Antarctic region. Both the Falkland Islands archipelago and South Georgia represent key areas that are host to significant avian biodiversity and the presence of HPAIV on these islands represents a significant risk to the susceptible bird populations. South Georgia is home to approximately 29 avian species which breed on the islands and is recognised as an 'Important Bird Area' by Birdlife International[53]. Therefore, any colony or population that comes under threat from an HPAIV outbreak on South Georgia may have direct impact upon the wider population of seabirds. Despite seabird colonies showing space partitioning between colonies[54], there is often a high degree of species overlap within colonies. Often this is due to the movement of nonbreeders or juvenile birds[55]. It is therefore, not unreasonable to suspect that birds on South Georgia may freely interact, which may aid the spread of disease, as has been documented previously[36,56]. Indeed, in the northern hemisphere it has been found that northern gannets (*Morus bassanus*) increased their interactions due to high levels of colony prospecting from surviving birds[56].

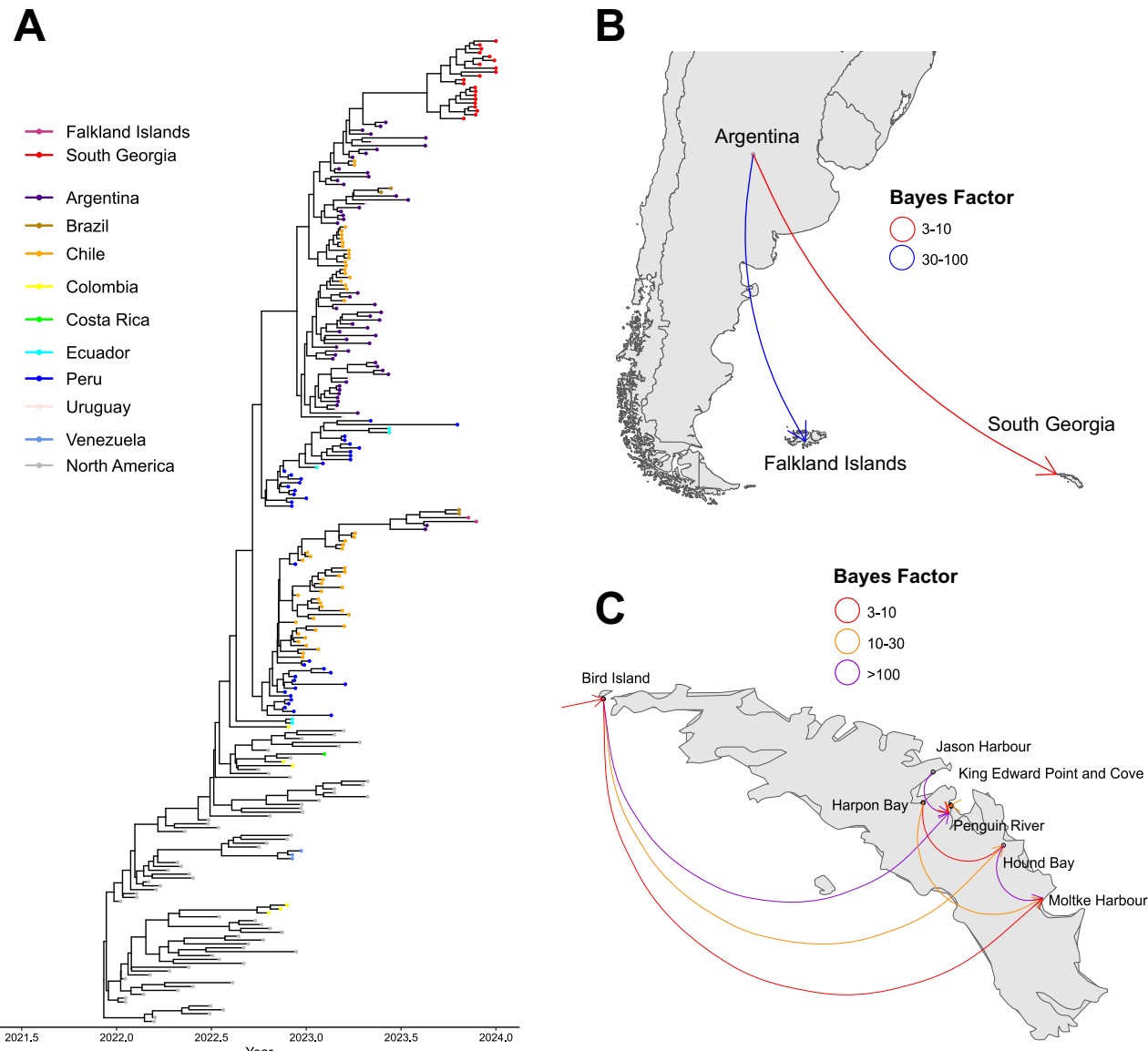

**Fig. 3 | H5N1 HPAIV transmission from the South American continent to Falkland Islands and South Georgia. A** Maximum clade credibility (MCC) phylogeny of the HA gene by BEAST analysis. The phylogeny is scaled by collection date of samples. Tip points are coloured by country of origin. **B** Analysis of South American, Falklands Islands and South Georgia H5N1 sequences suggests transmission from the mainland to the islands. BSSVS analysis is shown to describe potential transmission routes. Arrows are coloured according to relative strength, inferred using a bayes factor (BF), by which transmission events are supported. **C** Analysis of South Georgia H5N1 sequences suggests transmission between geographically related HPAIV detections. BSSVS analysis is shown to describe potential transmission routes across the island. Arrows are coloured according to relative strength, inferred using a bayes factor (BF), by which transmission events are supported.

The interlinkages between avian and mammalian species testing positive for H5N1, in closely linked ecosystems across the Antarctic region, means that there is potential for the virus to spread further and there have been reports of the virus being detected on mainland Antarctica[57,58] as well as causing low level mortality events in penguins (APHA, unpublished results)[58,59]. Circumpolar and trans-Pacific migrants such as grey-headed albatross[60], white-chinned petrel[61], southern giant petrels, and northern giant petrels (*Macronectes halli*) may facilitate this spread. Indeed, phylogeographic analysis has suggested a dynamic geneflow between southern Atlantic populations and Macquarie Island (an island in the southwestern Pacific Ocean, about halfway between New Zealand and Antarctica)[62], and as such the threat of transmission to New Zealand and Australasia must be considered.

From a mammalian infection standpoint there have been several reports globally of wild aquatic mammals, including seals, being infected with H5Nx HPAIV since 2020, where infection has been attributed to the predation of sick or dead infected birds[11,63,64]. Information to date suggests that HPAIV infection in seals often leads to a neurological presentation that may not result in viral shedding being detected through standard swab sampling activities[63]. This may explain the initial lack of influenza vRNA detection in elephant seal swab samples taken in this study, despite the consistency of clinical presentation seen in elephant seals with that reported elsewhere. However, follow up invasive sampling clearly demonstrated that swab samples from elephant seals were positive for vRNA demonstrating the utility of tissue sampling. Whilst weather conditions prevented a thorough sampling of

species from across the different islands/regions experiencing mortality events in seals, those achieved did include samples testing positive. Importantly, where mortality events continue to occur in cetacean species, the opportunity for infection of scavenging birds exists which may then continue the cycle of risk through the potential for disease events in birds and further shedding of virus into new environments. However, carcass removal, disposal and environmental clean-up is not an option. The Antarctic region is one of the most remote environments on earth and is the location of enormous breeding colonies of various avian species that may be susceptible, and succumb, to infection with HPAIV. Further, the potential for virus survival in this cold environment is increased and it may be that infectious virus remains for longer periods in carcasses preserved by the local climate. Local ecology of species could also influence the scale of impact throughout Antarctica. Although all species remain vulnerable to large scale infection events, it is possible that the density of animals may preclude some species from succumbing to viral spread[65]. For example, wandering albatross nest at low density (approximately 0.0022 nests per m$^2$)[66], which could limit spread between breeding individuals. However, non-breeding birds congregate in groups to display and dance[67] which may provide opportunities for disease spread. Similar ecological considerations must be made when considering burrow nesting species (such as white-chinned petrel, diving petrel [*Pelecanoides spp.*], and prion species [small petrels in the genera *Pachyptila* and *Halobaena*]), which nest in separated burrow systems and may limit spread. Penguins are also susceptible to HPAIV, and mortality has been observed following infection[68]. Penguin species nest in high densities (dependent upon species ranging between 0.25 and 1.7 nests per m$^2$)[69,70], and although limited impact has been seen to date, if HPAIV does start to impact more significantly on penguin colonies, it could show rapid infection and spread. However, the current situation does not suggest that significant mortality events will be seen in penguin species. The report of an outbreak in Gentoo penguin (*Pygoscelis papua*) chicks has not resulted in dissemination and mortality events in adult birds[59]. Activities within the region are ongoing to track mortality events and autonomous authorities are on high alert to signal the potential for incursions across the broader area although as suggested previously the proximity of affected species, both avian and mammalian, suggests that penguins may be less susceptible to this genotype of the virus.

Genomic analysis of the avian sequences obtained from South Georgia and the Falkland Islands suggested separate, distinct introductions of the B3.2 HPAIV genotype into the two locations. The B3.2 genotype emerged following co-infection (of a presumably avian host) with a viral descendant of the first GsGd HPAI virus detected in North America during 2021 (purportedly introduced from Europe) with a North American lineage virus. This genotype has been demonstrated to have been introduced into South America four times between October 2022 and March 2023[11]. Analysis of all available full-genome sequences from South America demonstrated that 95% (184 of 193) of H5N1 HPAIV sequences corresponded to this genotype. Given the close geographical proximity of South Georgia and the Falkland Islands to South America, and that wild bird species are known to migrate between the mainland and these islands, it is not surprising that the B3.2 was detected as the cause of the disease events. The phylogenetic analyses undertaken demonstrated that the viruses detected in South Georgia and the Falkland Islands shared common ancestors with those detected in mainland South America from mid-2023 and further analyses demonstrated that Argentina was the likely source of HPAIV. However, the long branch lengths observed across all gene segments suggests unsampled evolutionary ancestry. Further to this, analysis of the sequences obtained from the sampling sites on South Georgia suggests that after the initial introduction, there was spread of HPAIV

to multiple locations which is consistent with eye-witness accounts from the island.

The observation that all sequences generated from positive samples across these sampling activities conducted on South Georgia cluster closely within a monophyletic group, along with the BSSVS analysis demonstrates a singular introduction and that they do not constitute a divergent lineage. Further, the mammalian sequences cluster amongst the avian sequences demonstrating that they are not forming a separate evolutionary lineage. Contrary to this, the two sequences from the Falkland Islands do not form a monophyletic group and cluster with sequences from Argentina and Brazil. However, assessment of whether this is the result of multiple, separate introductions is limited due to the small number of sequences available at the time of writing. Further sampling activities within the Falkland Islands should be considered to further investigate the introduction of H5N1.

Critically, assessment of sequences derived from mammalian species did not indicate any adaptive mutations of increased risk to human populations. A single sequence from a single Southern elephant Seal sequence from Jason Harbour) contained the PB2 E627K mutation that is associated with adaptation to replication in mammalian species and a Kelp Gull sequence from Harpon Bay displayed PB2 D701N. Whilst the mutation at E627K is an early marker of mammalian adaptation, both further mutations in the PB2 gene and alterations to HA are required to define increased risk to human populations. No other mammalian or avian sequences investigated from South Georgia contained any of the other PB2 mutations of particular interest (i.e., T271A, K526R, Q591K or D701N). This contrasts with reports coming from sequences derived from mortality events in mammals across South America[11,71–73]. However, the two southern fulmar sequences obtained from the Falkland Islands contained Q591K and D701N mutations which match the mammal sequences from South America, potentially indicating different sources for introduction into the Antarctic region. The data generated here indicates that human populations on these islands are not at any increased risk from infection from these viruses. Further, there were no mutations that would affect the susceptibility to antivirals observed in the data generated in this study. Still, continual monitoring of the virus, especially where it appears to be the causative agents of mortality events in seals (and potentially further avian species where scavenging occurs on dead seals occurs), is critical to maintain an understanding of whether adaptive events may occur. Clearly, with continual infection events occurring in mammals globally it is of great importance to understand where mutations may represent a genuine zoonotic risk as well as understand where adaptations that have occurred in mammals become tolerated in avian species.

In conclusion, whilst the H5N1 B3.2 HPAIV has been translocated to the fragile ecosystem in the sub-Antarctic region, the current impact appears to be relatively minimal for avian species with minimal impact upon penguin populations being reported to date. Further whilst infection and significant mortality events have been observed in both fur and elephant seals, there remains no evidence of viral adaptation for enhanced infection of mammals and, consequently no increased risk to the human populations on the islands is predicted. Whilst this data supports a low impact on penguin species and little to no risk to humans, the global release of data restricts a fulsome interpretation here. It is of high interest to understand what impact repeat introduction events may be having in the region. Currently, there are only a limited number of sequences deposited in public databases from H5N1 HPAIV detections in South America during summer 2023, which limits interpretation. This factor, as with countless other studies highlights the importance of real-time global data sharing as a key tool in understanding the emergence and spread of these viruses. The current lack of publicly available data precludes a conclusive assessment of potential incursion routes. Multiple disciplines globally continue to

monitor the situation in Antarctica to see whether fears of ecological disaster in the region will be realised.

## Methods

### Sample collection

All samples were collected from carcasses found dead on the islands with the permissions of the Falkland Islands government and the government of South Georgia through collaboration with the British Antarctic survey. Sampling was only conducted where safe to do so and to minimize potential pathogen spread by fomite transfer to healthy animals. Data was collected by staff on the ground to define location, species and approximate age.

### Virological detection

On the Falkland Islands, initial diagnostic assessment of samples was undertaken at the KEMH Pathology and Food, Water & Environmental Laboratory utilising the QIAamp Viral RNA Mini Kit (Qiagen) and the Oasig OneStep RT-qPCR kit for H5N1 (Genesig). A preliminary diagnosis was made of avian influenza H5N1 infection. Following reports of increasing mortalities and the observation of disease consistent with HPAIV infection in avian and mammalian species in South Georgia, oropharyngeal (OP) and cloacal (C) swabs collected from birds were submitted to the Animal and Plant Health Agency (APHA)-Weybridge for laboratory virological investigation. Total nucleic acid was extracted from all samples[45] for testing by a suite of three AIV real-time reverse transcription polymerase chain reaction (rRT-PCR) assays consisting of the Matrix (M)-gene assay for generic influenza A virus detection[47,48], an assay for specific detection of HPAIV H5 clade 2.3.4.4b[45], and an N1-specific rRT-PCR to confirm the neuraminidase type[46]. All primers and probes used in these assays can be found in Supplementary Table 6A positive result was denoted in each case by a Cq value ≤ 36.0. The samples were also screened for avian paramyxovirus type 1 (APMV-1) by an rRT−PCR assay targeting the large polymerase (L) gene[74] where a positive result was denoted by a Cq value ≤ 37.0. All amplifications were carried out in an AriaMx qPCR System (Agilent, United Kingdom). The OP swabs, C swabs or brain material were used for virus isolation in 9- to 11-day-old specific pathogen-free (SPF) embryonated fowls' eggs (EFEs) according to the internationally recognised methods[75].

### Whole-genome sequencing and phylogenetic analysis

Wherever possible, sequence generation was undertaken on original clinical material. However, material derived following successful virus isolation in eggs was also utilised for sequence generation. For whole-genome sequence analysis, the extracted vRNA was converted to double-stranded cDNA and amplified using a one-step RT-PCR using SuperScript III One-Step RT-PCR kit (Thermo Fisher Scientific) as previously described[76,77]. The primers used can be found in Supplementary Table 6. PCR products were purified with Agencourt AMPure XP beads (Beckman Coultrer) prior to sequencing library preparation using the Native Barcoding Kit (Oxford Nanopore Technologies) and sequenced using a GridION Mk1 (Oxford Nanopore Technologies) according to manufacturer's instructions. Assembly of the influenza A viral genomes was performed using a custom in-house pipeline as described previously[8] but adapted for nanopore sequence reads. All influenza sequences generated and used in this study are available through the GISAID EpiFlu Database (https://www.gisaid.org). All H5N1 HPAIV clade 2.3.4.4b sequences available in the EpiFlu database between 1st September 2020 and 22nd January 2024 were downloaded to create a sequence dataset. As North America and Europe were over-represented in this dataset, these were sub-sampled to maintain representative sequences using PARNAS[78]. The remaining dataset was separated by segment and aligned using Mafft v7.520[79], and manually trimmed to the open-reading frame using Aliview version 1.26[80] The trimmed alignments were then used to a infer maximum-likelihood

phylogenetic tree using IQ-Tree version 2.2.3[81] along with ModelFinder were downloaded to create a sequence dataset. As North America and Europe were over-represented in this dataset, these were sub-sampled to maintain representative sequences using PARNAS[78]. The remaining dataset was separated by segment and aligned using Mafft v7.520[79], and manually trimmed to the open-reading frame using Aliview version 1.26[80]. The trimmed alignments were then used to a infer maximum-likelihood phylogenetic tree using IQ-Tree version 2.2.3[81] along with ModelFinder[8] and 1000 ultrafast bootstraps[82]. For the time-resolved phylogenetic analysis, all HA sequences available from South America, and representatives from North America were combined with the sequences from South Georgia and the Falkland Islands and used to infer phylogeny using BEAST version 1.10.4[83] with the BEAGLE library[84]. The Shapiro-Rambaut-Drummond-2006 (SRD06)[85] nucleotide substitution model was implemented with a four-category gamma distribution model of site-specific rate variation and separate partitions for[83] with the BEAGLE library[83,84]. The Shapiro-Rambaut-Drummond-2006 (SRD06)[85] nucleotide substitution model was implemented with a four-category gamma distribution model of site-specific rate variation and separate partitions for codon positions 1 plus 2 versus position 3 with the Hasegawa-Kishino-Yano (HKY) HKY substitution models on each with a strict clock and a coalescent GMRF Bayesian skyride tree prior. Three independent Markov Chain Monte Carlo (MCMC) runs were performed and combined using the Log Combiner tool in the BEAST package. Each chain consisted of 200,000,000 steps and was sampled every 20,000 steps, and the first 10% of samples were discarded as the burn-in. Discrete geographical transition events were reconstructed using a symmetric continuous-time Markov Chain model with an incorporated Bayesian stochastic search variable selection (BSSVS) to determine which transition rates sufficiently summarize connectivity[86]. SpreaD3 was used to determine the rates of transmission using a Bayes factor (BF) test. The BF represents the ratio of two competing statistical models, represented by their marginal likelihood, and, in this case, was used to determine the likelihood of transmission between geographical locations. The support of the BF for transmission was interpreted as described previously[87]). BF and representative transitions related to South Georgia and the Falkland Islands were visualised on maps in R (v4.3.2) using the packages GGPlot2[88], rnaturalearth[89], rnaturalearth data and sf[90]. Nucleotide identity between sequences was determined as described previously[82]. Sequences were genotyped according to the USDA schema, using the GenoFLU tool (https://github.com/USDA-VS/GenoFLU)[49].

### Reporting summary

Further information on research design is available in the Nature Portfolio Reporting Summary linked to this article.

## Data availability

All data is present in supplementary files or has been released on the NCBI database as indicated in the manuscript. Newly generated sequence data accession numbers are available in supplementary tables 1 and 2.

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

## Acknowledgements

We would like to thank Thomas Lewis and Vivian Coward from APHA for their help processing these samples and Carrie Gunn from BAS and Vicki Foster from the Government of South Georgia and the South Sandwich Islands. We are grateful for the support and collaboration provided by the Government of South Georgia and the South Sandwich Islands, who have helped to coordinate the response and guidance in response to HPAI at South Georgia. The authors would also like to acknowledge the originating and submitting laboratories of the sequences from GISAID's EpiFlu Database upon which this research is based, and analyses described in the text. A list of all sequences utilised from GISAID that

were not novel to this analysis are listed in Supplementary Table 7. All submitters of the data may be contacted directly via the GISAID website (https://www.gisaid.org). The analyses described in this work were conducted using the Scientific Computing Environment at the Animal and Plant Health Agency and the Nemo high-performance computing and data analytics platform at the Francis Crick Institute. This research received no external funding. The testing and generation of the viral sequences was funded by the Department for Environment, Food and Rural Affairs (Defra, UK) and the Devolved Administrations of Scotland and Wales, through the following programmes: SV3400, SV3032, SV3006, SE2213 and SE2227. This work was also supported by the Biotechnology and Biological Sciences Research Council (BBSRC) and Department for Environment, Food and Rural Affairs (Defra, UK) research initiative 'FluTrailMap' [grant number BB/Y007271/1] and the Medical Research Council (MRC) and Defra research initiative 'FluTrailMap-One Health' [grant number MR/Y03368X/1]. This work was also partially supported by KAPPA-FLU HORIZON-CL6-2022-FARM2FORK-02-03 (grant agreement No. 101084171) and Innovate UK (grant number 10085195).

## Author contributions

Conceptualisation: A.C.B., J.J., A.B., E.M.F., Z.F.; formal analysis: A.B., J.J., S.M.R., K.F., A.C.B., A.M.P.B.; investigation: S.M.R., M.F., J.G.L.J., D.d.S., F.B., M.B., R.H., A.M.P.B., J.P.D., B.M.; resources: A.C.B., J.J., I.H.B., Z.F., E.M.F.; writing—original draft, A.C.B., J.J., A.M.P.B., S.R.; writing—review and editing: A.C.B., J.J., A.P.M.B., A.B., Z.F., E.M.F., S.M.R., I.H.B., J.G.L.J., B.M. All authors have read and agreed to the final version of the manuscript.

## Competing interests

The authors declare no competing interests.
