## [Peer Review File · Nature Communications]

Detection and spread of high pathogenicity avian influenza virus H5N1 in the Antarctic regionREVIEWER COMMENTS

Reviewer #1 (Remarks to the Author):

NC-466298 Review

Summary:

The manuscript by Bennison et al. entitled "Detection and spread of high pathogenicity avian influenza virus H5N1 in the Antarctic Region" describes several avian species infected with HPAI H5N1 that were primarily sampled in South Georgia Island (within the Antarctic Polar Front), and secondarily in the Falkland Islands (adjacent to the Antarctic Polar Front) between October and November 2023. In total, 49 samples, from 24 individuals representing 5 species of birds, and 6 samples, from 3 individuals representing 1 species of mammal, were sampled. All mammalian samples were negative for HPAI H5N1. Most avian samples (~3/4) were positive for HPAI H5N1, but virus isolation was successful in only 4 cases, and genomes were reported to public repositories also in only 4 cases. However, the genomes reported to public repositories are not derived from the same samples where virus isolation was successful. Based on these data, and using epidemiological meta-data as support, the authors describe emergence of HPAI H5N1 within the Antarctic region, and then speculate on possible spread forward.

Major comments:

The identification of HPAI H5N1 within the Antarctic region is, in and of itself, compelling enough to warrant publication in Nature Communications, provided that the data are presented clearly and completely in order to communicate the importance and relevance of the findings. Several of the authors in this manuscript have extensive experience in the influenza field and knowledge of how to conduct these types of analyses properly. It is because of this experience that this reviewer was surprised to find that the manuscript was not particularly well written, and that many key data were not included, or partially described, or insufficiently explained. Often, shortcomings were not discussed/addressed. A basic example: the summary list I provide above, which includes how many samples, how many species, how many positives, etc., were tested is something I have had to count myself from the Supplementary Data provided. These details are super important, yet the authors never clearly state them in the main text. Additionally, key information, like the CT values for PCR-positive samples, or the coverage length and depth for the NGS data deposited in public repositories is missing. The methods are often quite thin, and not enough for a reader to be able to fully replicate their work. There is no photographic record of the animals (much less mention of pathology reports), or explanations for why virus isolation was only possible in 4 cases, or for why only 4 genomes were generated. In fact, there is no explanation for why the 4 samples where virus isolation was possible, were also not sequenced (that's something that should be done, and the pros and cons of sequencing a primary sample vs. an isolate should be addressed). Because of these reasons, this reviewer considers that the paper in its current form is not suitable for publication in Nature Communications unless it undergoes major revisions, that all the missing data are included, and that the viral isolates that were successful are sequenced and subsequently included in the phylogenetic analysis (which even the authors admit reveals massive under sampling. So, if you already know your data is under sampled, why would you not take advantage of all the samples you have on hand and try to reduce your under sampling?

Minor Comments:

Line 28: comma missing after "Falkland Islands"

Line 30: check and be consistent with your plurals... I think you mean "fur seals"

Line 32: your wording is repetitive. You just mentioned "South Georgia" in the sentence above... rewrite for consistency and flow.

Line 42: reference if missing for "Europe"

Line 52: careful with reference #13. That was not primary literature and it cited a BioRx paper that subsequently retracted a lot of the data they shared in their original submission to BioRx.

Line 56: "several islands are located within the Antarctic region..." a map early on would be helpful (a clear one, that includes which islands are within the Antarctic Polar Front (consider defining this area too) and which are not. The map you include later on is hard to see, so not very informative or helpful.

Line 96: First mention of Fig 1A, which is a map, but not a great one. As mentioned above, consider providing a complete map of Antarctica, including the Antarctic Polar Front, as well as the location of both S Georgia and Falklands relative to the polar front.

Line 100: Figure 1B doesn't match the statement "these individuals had died" or the symptoms described. Please provide photographic record of dead animals or change the statement that goes with Fig 1B.

Line 107 onwards: "...swab samples were collected from six found dead kelp gull and four found dead brown skua from Hound Bay, South Georgia in addition to four found dead brown skua from St Andrews, South Georgia (Figure 1B)" is a strange sentence. Repetitive and difficult to read. The antithesis of clarity. Check grammar and try to help your reader understand you easily, without having to re-read your sentences/paragraphs. "Six found dead kelp gull" sounds super strange. And then you repeat the same phrase structure twice more. Perhaps try to convey all of this repetitive info in a more useful and informative way? A timeline? In figure form, or in a table? Also, try to provide info in order. Right now in this paragraph you are all over the place: You first talk about sample types collected, then species, then location, then symptoms, then back to sample types, back to species, and back to locations. It makes for super inconvenient reading. And the entire time you refer to either Fig 1A or B, both of which are not helpful and not relevant to either sample types of species (only locations, which you can barely distinguish on your maps). Consider one figure, where A is a map (a clear, useful one that includes the polar front) and B is perhaps pics of the animals sampled.

Line 125: "local molecular testing"??? Can you be more specific? Can you explicitly say "RT-PCR", or whatever it is, so that the reader doesn't have to go to the materials and methods to dig up the info?

Line 133: first mention of RT-PCR, but references are not provided and reader sent to the supplementary materials to continue digging for important information. That is not acceptable, especially not for a manuscript submitted to a journal of this caliber.

Line 133: Supplementary Table 1 doesn't include CT values for positive samples (please include those), or metrics on the virus isolation (how long till you saw CPE? Or did you test in a different way? How did you confirm isolation?), or metrics on NGS (at the very least, discussion of coverage length and depth, with minimum coverage depth for all complete genome segments. These data are also important because they may help explain some of your other observations, like "infectious virus could not be isolated" (and yet your Supplementary Table lists 4 positive isolations, so where's the mistake?).

Line 134: You say "infectious virus could not be isolated" but Sup Table 1 shows 4 virus isolations in embryonated eggs. Not only that, but one of these is from a brain sample, which usually has much higher viral loads, so it makes total sense that you would be able to isolate from such a sample. So where's the discrepancy? There isn't one, only that you present your data in chronological order, which is not very helpful and in fact leads the reader to confusion. Careful that poor presentation of the data doesn't lead to misinterpretations, like did you do the experiments or not? "virus could not be isolated" is a very poor/limited description of what you did or what happened. Clearly state if you tried and failed? Or did you not try? Explain things clearly to your reader, so that they don't have to go fishing in the data to find support for your statements. Also consider that separate from virus isolation, you have other options in the lab, including partial sequencing of your PCR products, or better yet, full genome sequencing of the entire genome. You don't explicitly state any of this in the Results section. You should, as much as possible, use

concise language.

Line 140: "were negative in each assay" is super vague. Which assays are you referring to?

Line 142: "tested positive" is also super vague... explicitly name the assay.

Line 143: "in each assay" is more vagueness. What are you talking about? PCR and sequencing? PCR and isolation?

Line 146: "Three full genome sequences..." again, you are reporting your findings in chronological order, which is not helpful at all. You have a total of 4 genomes. You should be discussing these as a whole. Where did they come from, when, do you have full coverage, do you have partial coverage, what's the CT value of the samples, etc.

Line 148: "shared 99.86-100% nucleotide identity..." Can you provide some evidence? A figure? Can you back-up your statements with data? Or is the reader supposed to go to GenBank, download the sequence and do the comparison themselves? Same for Line 149 "a single sequence was also generated"... All of the important details/metrics are missing from the narration.

Line 151: "greater than 98%" this statement is also missing evidence.

Line 152: "combined" combined how? Provide details on which sequences were used... this is important to assess the quality of your analysis

Line 153: "assess genetic ancestry" Again, few details are provided on how the phylogenetic trees were constructed. Yes, they explain this in the materials and methods, but the results section needs enough info to stand on its own. The authors seem to rely/build heavily on the analysis of reference #10, but don't explain if what they did is a completely separate analysis that reaches the same conclusions, or a build-on. Authors talk about 131/140 sequences (Line 163), but the present study only generated 4 genomes, so its very easy for the reader to get lost. All of this could be clarified with a thorough re-write.

Line 170: "produced long branch lengths compared to South American sequences..." Meaning? Provide and interpretation please

Line 171: "discrete trait analysis based upon the country of origin was performed..." What does this mean? Also, how is Supplementary Fig 2 different from Fig 2 (other than they are colored differently?). What is this figure contributing to the analysis? Define "discrete trait analysis"

Line 187: "had" needs to be checked for grammar

Line 181: "being considered" avoid the passive voice

Line 192: "It is therefore, not unreasonable to suspect that birds on South Georgia may show high connectivity, which may aid the spread of disease, as has been documented previously³³, but also may be evidenced by the rapid collection of samples from different areas within South Georgia." This sentence makes little sense. Please reword.

Line 199: first definition of "connectivity." This This explanation/definition needs to be mentioned earlier, when the first bring up the topic of "connectivity"

Line 210: "despite the consistency of clinical presentation seen in elephant seals with that reported elsewhere." Can you think of any other explanations?

Line 214: "invasive sampling of avian and mammalian species remains challenging to undertake in areas where appropriate facilities are lacking" Verbose and repetitive. Consider editing down.

Line 226: "from rapid spread" Again, this is poorly phrased. You mean that the virus is spreading, but the sentence reads as if you are talking about a species spreading.

Line 249: "detected" check for grammar

Line 254: "during summer 2023" grammar

Line 256: "precludes a conclusive assessment of potential incursion routes substantially more difficult" makes no sense

Line 276: "separately pooled" what do you mean? Define pooled separately

Line 290: "all influenza sequences generated" there is no clarity on how many samples, how many individuals, how many species, how many genomes you have. One needs to go diving into sup materials to find out. You need to provide complete details, including NGS specs, coverage, CT values, etc.

Line 298: "bootstraps" full stop missing

Line 303: "mugration model" define or provide a reference

Line 358: "Discrete trait analysis" What does this mean? Figures 2 and Sup Figure 2: How are these different?

Reviewer #2 (Remarks to the Author):

NCOMMS-23-54471

Detection and spread of high pathogenicity avian influenza H5N1 in the Antarctic Region

Bennison and colleagues describe the relevant (and depressing) first observations of HPAI H5N1 in the Antarctic and sub-Antarctic regions of South Georgia and the Falkland Islands. The paper describes a highly relevant observation and is well written. Relevant data is provided for the described observations and conclusions.

I have only minor remarks to consider:

- Consider including Suppl table to the main manuscript. It is relevant for the reader to have a clear overview of samples collected and outcome of analyses on these samples.
- Line 41 and further: Consider shortly introducing the different genetic clades that have arisen since 1996 before jumping to mentioning clade 2.3.4.4b
- Line 42: it now seems clade 2.3.4.4b was first observed in 2021 in Europe. I do not think that is correct.
- Line 44: 'thousands' of outbreaks at poultry farms seems off.
- Line 100: not sure to what the reference to the figure actually refers to in this sentence.
- Line 170: consider explaining in a bit more detail what the long branches actually indicate/mean.
- Unfortunately there is no uniform classification system yet for reassortments/genotypes. IS there a labelling by EFSA for genotype B.3.2 as well? Genotype Herring_Gull/France like, is labeled BB by EFSA, but labeled differently by You et al who described the definition of B3.2. Due to lack of a uniform genotype labelling system, a schematic of the gene segment reassortment and their origin could be insightful
- Figure 1A: consider zooming out a bit more to make clearer which region it is

Reviewer #3 (Remarks to the Author):

I appreciate the opportunity to read and review the manuscript, 'Detection and spread of high pathogenicity avian influenza virus H5N1 in the Antarctic Region'. In this manuscript, the authors describe observations of sick and dead birds on South Georgia and the Falkland Island during October–December 2023 as well as diagnostic findings confirming the occurrence of highly pathogenic H5N1 avian influenza (HPAI) among a relatively small number of the diagnostic samples collected. This report signifies the first confirmed detection of HPAI in the Antarctic region and documents further geographic spread of the ongoing panzootic caused by infection with goose/Guangdong lineage viruses. The authors discuss the first detection of HPAI in the Antarctic region in the context of potential ecologic consequences.

In general, I found this manuscript to be well-written, to be based on apparently robust data, and to contain reasonable inference given the data summarized. I did not identify any obvious deficiencies pertaining to sampling design or diagnostic methods employed. My critiques are relatively minor and generally pertain to a perceived lack of clarity or precision which could introduce reader confusion. To this end, I've appended numerous, mostly editorial, suggestions below which I hope the authors may find helpful in the revision of their product.

I thank the authors for submitting their best work to Nature Communications.

Specific comments...

Lines 27, 34, and throughout: The authors might consider whether this manuscript describe the 'emergence' of HPAI in the Antarctic region or (geographic) spread thereto. Though an argument can certainly be made for the former, the latter seems irrefutable to me given the evidence.

Lines 64-67: My question here may be naïve, but are these 'resident' species, or rather, migratory species endemic to the Southern Ocean?

Line 81: What is meant by 'continental America'?

Line 82: The modifiers 'likely' and 'high' seem unnecessary to me. Also, 'dispersal' or 'spread' might be more precise than 'transmission' in this sentence when referencing spatial dissemination. Finally, this statement might be improved if it were made past tense.

Lines 100-102: I presume Bird Island is considered to be part of South Georgia or a broader South Georgia island group. It would be helpful to your reader to clarify (e.g., Bird Island and South Georgia appear, at times, to be used interchangeably throughout the manuscript). Also, was this mortality on Bird Island also among brown skuas or other species? It is somewhat unclear as written.

Line 118: Perhaps events might be described as 'occurring' rather than 'emerging'.

Line 121: The article 'a' should be added before 'Falkland steamer duck'.

Lines 95-143: You might consider referencing Supplementary Table 1 periodically throughout this text to help out your reader(s).

Lines 158-163: This same information is conveyed almost verbatim in the Discussion (lines 239-246). Please include only once to avoid repetition. Also, please consider using 'dispersal' or 'spread' rather than 'transmission' to refer to spatial dissemination.

Line 166: Perhaps 'provided evidence for' rather than 'demonstrated'.

Lines 170-173: I personally found this analysis to be weak given probable sampling/reporting biases (e.g., the dearth of sequences from Argentina). I note that the authors do not reference this analysis in the Discussion. Given inherent uncertainty in this analysis and the lack of contribution to the overall narrative, I suggest that the authors consider omitting from their product.

Line 178: The authors might clarify that they are referencing the Antarctic mainland here.

Line 187: Consider using 'susceptible bird populations' rather than 'the populations of sensitive bird species'.

194: The authors might consider ending their statement after reference 33, perhaps adding reference 49. I'm dubious that sample collection alone supports their claim (e.g., in the absence of genomic information or animal movement data) and the description of reference 49 doesn't add significant value to the narrative in my opinion.

Line 198-204: I don't follow the logic of this argument. I feel that the Subantarctic and Antarctic ecosystems are quite diverse and that pelagic birds may or may not be particularly likely to facilitate rapid HPAI spread. The authors should please 'take or leave' this viewpoint, but I'd encourage them to omit this particularly subjective text. Alternatively, they might revise/re-frame as I acknowledge that I may be misunderstanding the intent of messages.

Line 249: There is a grammatical issue here.

Figure 2: Given that HPAI detections in the Antarctic region and those in South America are consistently of genotype B3.2 viruses, phylogenetic analysis of complete concatenated viral genomes would potentially provide higher resolution inference than a tree generated using only data for the H5 HA gene segment. Also, the authors might consider using 'dispersal' or 'spread' rather than 'transmission' in the legend.

Response to Reviewers – NCOMMS-23-54471 –

Detection and spread of high pathogenicity avian influenza H5N1 in the Antarctic Region

We thank the reviewers for their fulsome assessment of the work we submitted. Alongside amending the manuscript to satisfy the reviewer's comments we have also added a considerable amount of data to further define the situation in the area. As such, the manuscript has been thoroughly amended with highlighting throughout to demonstrate where changes have been made. Below we offer a point-by-point rebuttal to comments on the original article with our responses being *italicised and in bold* font for clarity.

Reviewer #1 (Remarks to the Author):

The manuscript by Bennison et al. entitled "Detection and spread of high pathogenicity avian influenza virus H5N1 in the Antarctic Region" describes several avian species infected with HPAI H5N1 that were primarily sampled in South Georgia Island (within the Antarctic Polar Front), and secondarily in the Falkland Islands (adjacently external to the Antarctic Polar Front) between October and November 2023. In total, 49 samples, from 24 individuals representing 5 species of birds, and 6 samples, from 3 individuals representing 1 species of mammal, were sampled. All mammalian samples were negative for HPAI H5N1. Most avian samples (~3/4) were positive for HPAI H5N1, but virus isolation was successful in only 4 cases, and genomes were reported to public repositories also in only 4 cases. However, the genomes reported to public repositories are not derived from the same samples where virus isolation was successful. Based on these data, and using epidemiological meta-data as support, the authors describe emergence of HPAI H5N1 within the Antarctic region, and then speculate on possible spread forward.

Major comments:

The identification of HPAI H5N1 within the Antarctic region is, in and of itself, compelling enough to warrant publication in Nature Communications, provided that the data are presented clearly and completely in order to communicate the importance and relevance of the findings. Several of the authors in this manuscript have extensive experience in the influenza field and knowledge of how to conduct these types of analyses properly. It is because of this experience that this reviewer was surprised to find that the manuscript was not particularly well written, and that many key data were not included, or partially described, or insufficiently explained. Often, shortcomings were not discussed/addressed. A basic example: the summary list I provide above, which includes how many samples, how many species, how many positives, etc., were tested is something I have had to count myself from the Supplementary Data provided.

These details are super important, yet the authors never clearly state them in the main text. Additionally, key information, like the CT values for PCR-positive samples (Added line XXX/Table XXX),

We have included this information

or the coverage length and depth for the NGS data deposited in public repositories is missing.

Because of the limitations of the samples we have, we have only been able to include consensus sequence. As such depth of NGS data has little relevance and so has been omitted.

The methods are often quite thin, and not enough for a reader to be able to fully replicate their work.
We have expanded the methods

There is no photographic record of the animals (much less mention of pathology reports),
We have included photographic images in Figures 1B and 1C. We cannot add pathological details as there aren't trained pathologists on the islands and we haven't been able to ship carcasses because of the location and shipping methods.

or explanations for why virus isolation was only possible in 4 cases, or for why only 4 genomes were generated.

We have included this information

In fact, there is no explanation for why the 4 samples where virus isolation was possible, were also not sequenced (that's something that should be done, and the pros and cons of sequencing a primary sample vs. an isolate should be addressed).

We have included this information

Because of these reasons, this reviewer considers that the paper in its current form is not suitable for publication in Nature Communications unless it undergoes major revisions, that all the missing data are included, and that the viral isolates that were successful are sequenced and subsequently included in the phylogenetic analysis (which even the authors admit reveals massive under sampling. So, if you already know your data is under sampled, why would you not take advantage of all the samples you have on hand and try to reduce your under sampling?

Minor Comments:

1. Line 28: comma missing after “Falkland Islands”

We have added a comma as suggested.

2. Line 30: check and be consistent with your plurals... I think you mean “fur seals”

This has been amended as requested.

3. Line 32: your wording is repetitive. You just mentioned “South Georgia” in the sentence above... rewrite for consistency and flow.

This has been amended as requested.

4. Line 42: reference if missing for “Europe”

We have added a suitable reference.

5. Line 52: careful with reference #13. That was not primary literature and it cited a BioRx paper that subsequently retracted a lot of the data they shared in their original submission to BioRx.

Thank you for this very helpful observation. We hadn't noticed the inaccuracies of this reference and have replaced it with a more recent article.

6. Line 56: “several islands are located within the Antarctic region...” a map early on would be helpful (a clear one, that includes which islands are within the Antarctic Polar Front (consider defining this area too) and which are not. The map you include later on is hard to see, so not very informative or helpful.

We have replaced the map to have the information as requested.

7. Line 96: First mention of Fig 1A, which is a map, but not a great one. As mentioned above, consider providing a complete map of Antarctica, including the Antarctic Polar Front, as well as the location of both S Georgia and Falklands relative to the polar front.

We have amended and included this further figure as a part of a panel with the figure requested in point 6. It is now Figure 1B.

8. Line 100: Figure 1B doesn't match the statement "these individuals had died" or the symptoms described. Please provide photographic record of dead animals or change the statement that goes with Fig 1B.

We have decided not to show images of dead animals as the local governments would rather such images don't form part of the manuscript, and certainly, what does an image of a dead animal actually add from a scientific perspective? We have altered the statement.

9. Line 107 onwards: “...swab samples were collected from six found dead kelp gull and four found dead brown skua from Hound Bay, South Georgia in addition to four found dead brown skua from St Andrews, South Georgia (Figure 1B)” is a strange sentence. Repetitive and difficult to read. The antithesis of clarity. Check grammar and try to help your reader understand you easily, without having to re-read your sentences/paragraphs. "Six found dead kelp gull" sounds super strange. And then you repeat the same phrase structure twice more. Perhaps try to convey all of this repetitive info in a more useful and informative way? A timeline? In figure form, or in a table? Also, try to provide info in order. Right now in this paragraph you are all over the place: You first talk about sample types collected, then species, then location, then symptoms, then back to sample types, back to species, and back to locations. It makes for super inconvenient reading. And the entire time you refer to either Fig 1A or B, both of which are not helpful and not relevant to either sample types or species (only locations, which you can barely distinguish on your maps). Consider one figure, where A is a map (a clear, useful one that includes the polar front) and B is perhaps pics of the animals sampled.

We agree with the reviewer and have rationalised the text with reference to Figure 1B, Table 1 and the full data set in Table S1.

10. Line 125: “local molecular testing”??? Can you be more specific? Can you explicitly say “RT-PCR”, or whatever it is, so that the reader doesn't have to go to the materials and methods to dig up the info?

We have amended the text to avoid this necessity.

11. Line 133: first mention of RT-PCR, but references are not provided and reader sent to the supplementary materials to continue digging for important information. That is not acceptable, especially not for a manuscript submitted to a journal of this caliber.

We have added refs as suggested would like to state that the molecular methods are clearly stated in the methods section.

12. Line 133: Supplementary Table 1 doesn't include CT values for positive samples (please include those), or metrics on the virus isolation (how long till you saw CPE? Or did you test in a different way? How did you confirm isolation?), or metrics on NGS (at the very least, discussion of coverage length and depth, with minimum coverage depth for all complete genome segments. These data are also important because they may help explain some of your other observations, like "infectious virus could not be isolated" (and yet your Supplementary Table lists 4 positive isolations, so where's the mistake?).

We have updated the table that now includes Ct values and have included positive and negative samples alongside where we have successfully isolated virus and instances where we have generated sequence from clinical material. The process of taking and shipping samples halfway across the globe means that even where strong PCR positivity is seen, live virus may not have been recovered as infectious material may have degraded in transit. As such we have undertaken direct sequencing of clinical material and attempted isolation in eggs as per WOAHA guidelines, but with limited success with the latter. From the perspective of genomic data, we limited the analysis to those where we could generate complete genomes and have assessed consensus data. As such coverage depth is of limited value.

13. Line 134: You say "infectious virus could not be isolated" but Sup Table 1 shows 4 virus isolations in embryonated eggs. Not only that, but one of these is from a brain sample, which usually has much higher viral loads, so it makes total sense that you would be able to isolate from such a sample. So where's the discrepancy? There isn't one, only that you present your data in chronological order, which is not very helpful and in fact leads the reader to confusion. Careful that poor presentation of the data doesn't lead to misinterpretations, like did you do the experiments or not? "virus could not be isolated" is a very poor/limited description of what you did or what happened. Clearly state if you tried and failed? Or did you not try? Explain things clearly to your reader, so that they don't have to go fishing in the data to find support for your statements. Also consider that separate from virus isolation, you have other options in the lab, including partial sequencing of your PCR products, or better yet, full genome sequencing of the entire genome. You don't explicitly state any of this in the Results section. You should, as much as possible, use concise language.

Thank you for this comment although we feel that this is a slight misinterpretation of the existing text. Regardless we have amended the point for clarity.

14. Line 140: "were negative in each assay" is super vague. Which assays are you referring to?
We have amended the text to reflect this.

15. Line 142: "tested positive" is also super vague... explicitly name the assay.
We have amended as requested.

16. Line 143: "in each assay" is more vagueness. What are you talking about? PCR and sequencing? PCR and isolation?
We have amended as requested.

17. Line 146: "Three full genome sequences..." again, you are reporting your findings in chronological order, which is not helpful at all. You have a total of 4 genomes. You should be discussing these as a whole. Where did they come from, when, do you have full coverage, do you have partial coverage, what's the CT value of the samples, etc.

In the amended version we have now included a total of 23 genomes (20x avian derived HPAIV genomes; 3x mammalian derived HPAIV genomes) and have amended this text as a result.

18. Line 148: “shared 99.86-100% nucleotide identity...” Can you provide some evidence? A figure? Can you back-up your statements with data? Or is the reader supposed to go to GenBank, download the sequence and do the comparison themselves? Same for Line 149 “a single sequence was also generated”... All of the important details/metrics are missing from the narration.

We have amended as suggested and have included a further Supplementary Table S3 with this information.

19. Line 151: “greater than 98%” this statement is also missing evidence.

See Supplementary Table S3 as previous comment.

20. Line 152: “combined” combined how? Provide details on which sequences were used... this is important to assess the quality of your analysis

We have amended the text for clarity.

21. Line 153: “assess genetic ancestry” Again, few details are provided on how the phylogenetic trees were constructed. Yes, they explain this in the materials and methods, but the results section needs enough info to stand on its own. The authors seem to rely/build heavily on the analysis of reference #10, but don't explain if what they did is a completely separate analysis that reaches the same conclusions, or a build-on. Authors talk about 131/140 sequences (Line 163), but the present study only generated 4 genomes, so its very easy for the reader to get lost. All of this could be clarified with a thorough re-write.

With the inclusion of further genomes we have edited the text to satisfy this comment

22. Line 170: “produced long branch lengths compared to South American sequences...” Meaning? Provide and interpretation please

We have added some text to clarify.

23. Line 171: “discrete trait analysis based upon the country of origin was performed...” What does this mean? Also, how is Supplementary Fig 2 different from Fig 2 (other than they are colored differently?). What is this figure contributing to the analysis? Define “discrete trait analysis”

With the inclusion of additional sequences, we have revised this analysis to infer the likely source of HPAIV from South America and further spread within South Georgia.

24. Line 187: “had” needs to be checked for grammar

We have amended for grammar.

25. Line 181: “being considered” avoid the passive voice

This has been amended.

26. Line 192: “It is therefore, not unreasonable to suspect that birds on South Georgia may show high connectivity, which may aid the spread of disease, as has been documented previously³³, but also may be evidenced by the rapid collection of samples from different areas within South Georgia.” This sentence makes little sense. Please reword.

This has been reworded.

27. Line 199: first definition of “connectivity.” This This explanation/definition needs to be mentioned earlier, when the first bring up the topic of "connectivity"

This has been reworded.

28. Line 210: “despite the consistency of clinical presentation seen in elephant seals with that reported elsewhere.” Can you think of any other explanations?

We have added additional samples and as such this comment is no longer relevant to the reported text.

29. Line 214: “invasive sampling of avian and mammalian species remains challenging to undertake in areas where appropriate facilities are lacking” Verbose and repetitive. Consider editing down.

Amended as requested.

30. Line 226: “from rapid spread” Again, this is poorly phrased. You mean that the virus is spreading, but the sentence reads as if you are talking about a species spreading.

This point has been resolved.

31. Line 249: “detected” check for grammar

This point has been resolved.

32. Line 254: “during summer 2023” grammar

This point has been resolved.

33. Line 256: “precludes a conclusive assessment of potential incursion routes substantially more difficult” makes no sense

This point has been resolved.

34. Line 276: “separately pooled” what do you mean? Define pooled separately

This point has been resolved.

35. Line 290: “all influenza sequences generated” there is no clarity on how many samples, how many individuals, how many species, how many genomes you have. One needs to go diving into sup materials to find out. You need to provide complete details, including NGS specs, coverage, CT values, etc.

This information is detailed in Table 1 and Supplementary Tables S1 and S2.

36. Line 298: “bootstraps” full stop missing.

This point has been resolved.

37. Line 303: “mugration model” define or provide a reference.

This point has been resolved.

38. Line 358: “Discrete trait analysis” What does this mean? Figures 2 and Sup Figure 2: How are these different?

With the inclusion of additional sequences from South Georgia and the Falkland Islands, we have revised this analysis using BEAST to infer the likely source of HPAIV from South America, and all text and figures related to this has been updated.

Reviewer #2 (Remarks to the Author):

Bennison and colleagues describe the relevant (and depressing) first observations of HPAI H5N1 in the Antarctic and sub-Antarctic regions of South Georgia and the Falkland Islands. The paper describes a highly relevant observation and is well written. Relevant data is provided for the described observations and conclusions.

We thank the reviewer for these kind comments.

I have only minor remarks to consider:

39. Consider including Suppl table to the main manuscript. It is relevant for the reader to have a clear overview of samples collected and outcome of analyses on these samples.

We have made included a summary table into the main text following this insightful suggestion.

40. Line 41 and further: Consider shortly introducing the different genetic clades that have arisen since 1996 before jumping to mentioning clade 2.3.4.4b

This has been amended.

41. Line 42: it now seems clade 2.3.4.4b was first observed in 2021 in Europe. I do not think that is correct.

This has been amended.

42. Line 44: ‘thousands’ of outbreaks at poultry farms seems off.

This has been amended.

43. Line 100: not sure to what the reference to the figure actually refers to in this sentence.

This has been amended.

44. Line 170: consider explaining in a bit more detail what the long branches actually indicate/mean.

Long branch lengths tend to suggest unsampled ancestry. We have noted this in the manuscript.

45. Unfortunately there is no uniform classification system yet for reassortments/genotypes. IS there a labelling by EFSA for genotype B.3.2 as well? Genotype Herring_Gull/France like, is labeled BB by EFSA, but labeled differently by You et al who described the definition of B3.2. Due to lack of a uniform genotype labelling system, a schematic of the gene segment reassortment and their origin could be insightful

Thank you for this comment, and we agree the absence of a harmonised global nomenclature system for the current H5 clade 2.3.4.4bs is much needed to help clarification regarding communication of these genotypes. Unfortunately, EFSA only classifies and names genotypes that have been observed in Europe, and the B3.2 genotype has not been detected in Europe to date, and there is therefore no corresponding EFSA name for this genotype.

46. Figure 1A: consider zooming out a bit more to make clearer which region it is.

We have made a new figure 1A, 1B and 1C.

Reviewer #3 (Remarks to the Author):

I appreciate the opportunity to read and review the manuscript, ‘Detection and spread of high pathogenicity avian influenza virus H5N1 in the Antarctic Region’. In this manuscript, the authors describe observations of sick and dead birds on South Georgia and the Falkland Island during October–December 2023 as well as diagnostic findings confirming the occurrence of highly pathogenic H5N1 avian influenza (HPAI) among a relatively small number of the diagnostic samples collected. This report signifies the first confirmed detection of HPAI in the Antarctic region and documents further geographic spread of the ongoing panzootic caused by infection with goose/Guangdong lineage viruses. The authors discuss the first detection of HPAI in the Antarctic region in the context of potential ecologic consequences.

In general, I found this manuscript to be well-written, to be based on apparently robust data, and to contain reasonable inference given the data summarized. I did not identify any obvious deficiencies pertaining to sampling design or diagnostic methods employed. My critiques are relatively minor and generally pertain to a perceived lack of clarity or precision which could introduce reader confusion. To this end, I’ve appended numerous, mostly editorial, suggestions below which I hope the authors may find helpful in the revision of their product.

I thank the authors for submitting their best work to Nature Communications.

We thank the reviewer for these kind comments.

Specific comments...

47. Lines 27, 34, and throughout: The authors might consider whether this manuscript describe the ‘emergence’ of HPAI in the Antarctic region or (geographic) spread thereto. Though an argument can certainly be made for the former, the latter seems irrefutable to me given the evidence.

We have been through the manuscript and have stated the correct descriptor as necessary.

48. Lines 64-67: My question here may be naïve, but are these ‘resident’ species, or rather, migratory species endemic to the Southern Ocean?

We have altered the text to make this clearer.

49. Line 81: What is meant by ‘continental America’?

We have altered the text to state South America.

50. Line 82: The modifiers ‘likely’ and ‘high’ seem unnecessary to me. Also, ‘dispersal’ or ‘spread’ might be more precise than ‘transmission’ in this sentence when referencing spatial dissemination. Finally, this statement might be improved if it were made past tense.

We have altered the text to state ‘elevated risk’ rather than the previous modifiers.

51. Lines 100-102: I presume Bird Island is considered to be part of South Georgia or a broader South Georgia island group. It would be helpful to your reader to clarify (e.g., Bird Island and South Georgia appear, at times, to be used interchangeably throughout the manuscript). Also, was this mortality on Bird Island also among brown skuas or other species? It is somewhat unclear as written.

We have altered the text to clarify the naming of the geographical locations and affected species.

52. Line 118: Perhaps events might be described as ‘occurring’ rather than ‘emerging’.

We have altered the text as suggested.

53. Line 121: The article ‘a’ should be added before ‘Falkland steamer duck’.

We have altered the text as suggested.

54. Lines 95-143: You might consider referencing Supplementary Table 1 periodically throughout this text to help out your reader(s).

We have altered the text as suggested.

55. Lines 158-163: This same information is conveyed almost verbatim in the Discussion (lines 239-246). Please include only once to avoid repetition. Also, please consider using ‘dispersal’ or ‘spread’ rather than ‘transmission’ to refer to spatial dissemination.

We have altered the text as suggested to avoid repetition.

56. Line 166: Perhaps ‘provided evidence for’ rather than ‘demonstrated’.

We have altered the text as suggested.

57. Lines 170-173: I personally found this analysis to be weak given probable sampling/reporting biases (e.g., the dearth of sequences from Argentina). I note that the authors do not reference this analysis in the Discussion. Given inherent uncertainty in this analysis and the lack of contribution to the overall narrative, I suggest that the authors consider omitting from their product.

We have updated the whole manuscript with further data from more recent sampling activities and hence have significantly improved this section in line with the release of sequence from other countries and a new phylogenetic assessment.

58. Line 178: The authors might clarify that they are referencing the Antarctic mainland here.

We have altered the text as suggested.

59. Line 187: Consider using ‘susceptible bird populations’ rather than ‘the populations of sensitive bird species’.

We have altered the text as suggested.

60. 194: The authors might consider ending their statement after reference 33, perhaps adding reference 49. I’m dubious that sample collection alone supports their claim (e.g., in the absence of genomic information or animal movement data) and the description of reference 49 doesn’t add significant value to the narrative in my opinion.

We have altered the text as suggested.

61. Line 198-204: I don’t follow the logic of this argument. I feel that the Subantarctic and Antarctic ecosystems are quite diverse and that pelagic birds may or may not be particularly likely to facilitate rapid HPAI spread. The authors should please ‘take or leave’ this viewpoint, but I’d encourage them to omit this particularly subjective text. Alternatively, they might revise/re-frame as I acknowledge that I may be misunderstanding the intent of messages.

We have balanced our response to state that close geographical linkages may be a factor in further spread.

62. Line 249: There is a grammatical issue here.

We have altered the text as suggested.

63. Figure 2: Given that HPAI detections in the Antarctic region and those in South America are consistently of genotype B3.2 viruses, phylogenetic analysis of complete concatenated viral genomes would potentially provide higher resolution inference than a tree generated using only data for the H5 HA gene segment. Also, the authors might consider using ‘dispersal’ or ‘spread’ rather than ‘transmission’ in the legend.

Thank you for this helpful comment. At the time of the original submission, whilst there was a good amount of full genome sequences from South America (~130), there were more HA sequences available there were a number of gaps in the sequences that were publicly available from South America (~170). Importantly, there was a single sequence from Argentina available that only included the HA gene and we therefore sought to use the HA gene alone for this analysis to ensure we had the most geographical representation as possible. Since then, a substantial number of sequences from Argentina, the majority of which are complete genomes have been made available. However, the number of HA sequences from South America (~230) still outweighs the number of complete genomes (~190), therefore using the HA only still allows us to maximise the amount of genetic and geographical diversity represented in the analyses. These newly released South American sequences, along with additional sequences from South Georgia have been included in our updated analyses.

REVIEWER COMMENTS

Reviewer #1 (Remarks to the Author):

I guess many of my comments have been addressed (especially the minor ones), though in all honesty it has been hard to assess this, as the responses are mostly "the comments have been addressed" or "we have amended the text," but when one goes to the text it is very hard/impossible to find the change since the new manuscript is completely different from the old one and there is no explicit mention in the response to reviewers how/where it is that they have "addressed the comments" or "amended the text." Also, the new version of the manuscript does not include track changes, which makes the work of a conscientious reviewer who provided honest feedback in good faith harder than it should be. I will say that this version of the manuscript is MUCH improved over the previous one (congrats and thank you!), and I was hoping that I could say yes, go ahead and publish as is, as the discovery of HPAI H5N1 in Antarctica is super compelling on its own. However, there are still some remaining points that I think are important and where I emphatically disagree with the authors.

Line 37: They say "Critically, genetic assessment of sequences from mammalian species demonstrates no increased risk to human populations." And again later in Line 329: "Critically, assessment of sequences derived from mammalian species did not indicate any adaptive mutations or increased risk to human populations. A single sequence from a single Southern elephant Seal sequence from Jason Harbour) contained the PB2 E627K mutation that is associated with adaptation to replication in mammalian species..." My issue is: how can you claim no increased risk to human populations when you find one of *THE* most important markers of mammalian host adaptation/pathogenicity, which is PB2 E627K present in a mammal?! Is it because you only find it in one sample? Isn't it also true that you recognize that you are vastly under-sampled? Your trees show it! Not to mention issues with your sample collection/testing. It doesn't escape me that working in Antarctica is incredibly difficult, and the logistics perhaps sometimes insurmountable. But the explanations provided to rebut the good critical feedback that was given during the first round of reviews are disappointing. For example, Line 258: "Information to date suggests that HPAIV infection in seals often leads to a neurological presentation with infrequent detection of viral material being detected through standard swab sampling activities. This may explain the initial lack of influenza vRNA detection in elephant seal swab samples taken in this study." Claiming "infrequent detection" in swabs is misleading. What is true is that the brains of infected animals have higher viral loads than other tissues, especially when they have encephalitis, but that does not mean you cannot pull this from swabs (or other tissues), so a more thoughtful and honest discussion of why you didn't or couldn't would be appreciated.

Another point: I asked you to include minimum coverage depth for your sequences, but you claim that this has "little relevance" since you are reporting consensus genomes. I completely disagree that coverage depth is of little relevance, even for consensus genomes. Yes, we often translate NGS data into consensus genomes, particularly in a situation like this one, where one is not looking at intra-host viral diversity, but a consensus genome with a coverage of 1x is completely different from a consensus genome with coverage of 10,000x. Coverage depth is a measure of certainty, of probability (in a sense) that the sequence you are reporting is correct, so for you to dismiss this as "not relevant" is not appropriate. Asking for coverage depth is the least I could ask you for. I could be asking you to deposit your raw data into public repositories for others to scrutinize, and yet, because of the gravity of the situation I am asking for simple information that you already have. Why do you not want to show it? Is your data not solid? Or is it that you don't trust it? Or do you not find it compelling enough? I would suspect that it is because there are parts of the genomes where your coverage depth is zero –if so, this should be discussed, not hidden. We all generate data where some parts of the genome have low or zero coverage depth. I went to GISAID to look for the sequences and they were hard to find (you also don't provide accession numbers), so along with the fact that you claim "no increased risk to humans" despite having E627K and being under-sampled worries me tremendously. Finally, line 340: "should the situation change..." I would argue the situation is already changing, only that you are catching it early and therefore you don't have all the evidence clearly displayed in front of you (just a few previews). And rather than seeing the situation for what it is (you have E627K!!!! how much more compelling can it get?!?!?!), you don't grasp its potential severity, and the potentially critical finding of E627K in a mammal. Yes, it is true you only find it only in one sample thus far, but if it is in one sample already, it is a matter of time before it is found in others.

Reviewer #3 (Remarks to the Author):

I appreciate the opportunity to review the revised manuscript, 'Detection and spread of high pathogenicity avian influenza virus H5N1 in the Antarctic Region' submitted to Nature Communications. The authors have substantially reworked their original submission in response to three reviewer critiques. Additionally, the authors have expanded the scope of the manuscript to include numerous additional detections of highly pathogenic avian influenza (HPAI) viruses in the Antarctic Region. They have also bolstered analyses through the inclusion of additional genetic information on HPAI viruses. The manuscript is much improved and also substantially altered as compared to the original submission.

From my perspective, the revised manuscript is conceptually sound and lacking any fatal flaws in reporting of information or interpretation. My assessment is that modest revisions would help to shore up several minor deficiencies within the narrative. I've appended (below) numerous comments towards this end. I thank the authors for their important contribution to Nature Communications and the broader scientific literature.

Lines 45-46: It is a gross over-simplification to state that GsGd HPAIV 'spread to other Asian countries'. Such viruses also spread through Europe, much of Africa, and into North America prior to 2021. Please revise this statement to make more congruent with the preceding and following statements.

Line 87: It is not explained how H11N2 viruses contrast with those of H4-H6 HA subtypes.

Line 89: Perhaps spread of low pathogenic avian influenza viruses to the Antarctic Region from both the Americas and elsewhere provides evidence on HPAI virus introduction risk? See prior comment.

Line 115: I believe 'preyed', rather than predated, may be the more appropriate term here.

Line 142: 6/17 mammalian carcasses are reported as HPAI virus positive in tables 1 and S2.

Line 178-180: There is a grammatical issue in this statement.

Line 188-189; I understand the 'full data set' to include only sequences generated in this study and not the representative sequences included in the maximum likelihood phylogenetic analyses as described throughout the preceding statements in the paragraph. As such, the statement explaining comparisons of nucleotide identity should probably be presented first with explanation of maximum likelihood phylogenetic analyses following thereafter.

Line 192: The full tree (Figure 2A) with complete tip labels including strain names and/or accession numbers should be provided to readers for transparency and so the analysis can be replicated. Such a figure could be provided as a supplemental file.

Line 226: There is a tense issue with the verb 'has not' (e.g., should be 'had' not given the report of HPAI in the Antarctic Region in this product).

Line 262-264: I'm not understanding the logic that invasive sampling demonstrated that swab samples were positive. Are the authors trying to convey that testing of tissue samples collected through necropsy was often consistent with diagnostic results for swab samples generated from the same individuals?

Lines 266-268, 289-295: I encourage the authors to focus discussion on information presented in their report, not unreported/unpublished/unreviewed observations that cannot be verified by readers.

Line 304: The phrasing 'the original H5N1' is not appropriate nor correct. Please rephrase along the lines of, 'genotype 3.2 emerged following co-infection (of a presumably avian host) with a viral descendant of the first GsGd HPAI virus detected in North America during 2021 (purportedly introduced from Europe) with a North American lineage virus'.

Bennison et al., 2024- Rebuttal to revision

Again we thank the reviewer's for their insightful comments on our manuscript. A rebuttal to each point is listed below and a line number added for ease of checking against the amended manuscript.

I guess many of my comments have been addressed (especially the minor ones), though in all honesty it has been hard to assess this, as the responses are mostly "the comments have been addressed" or "we have amended the text," but when one goes to the text it is very hard/impossible to find the change since the new manuscript is completely different from the old one and there is no explicit mention in the response to reviewers how/where it is that they have "addressed the comments" or "amended the text." Also, the new version of the manuscript does not include track changes, which makes the work of a conscientious reviewer who provided honest feedback in good faith harder than it should be. I will say that this version of the manuscript is MUCH improved over the previous one (congrats and thank you!), and I was hoping that I could say yes, go ahead and publish as is, as the discovery of HPAI H5N1 in Antarctica is super compelling on its own. However, there are still some remaining points that I think are important and where I emphatically disagree with the authors.

We thank this reviewer for their critical appraisal of the revised manuscript and have written a response to each comment below. We empathise 100% with the comments around tracking changes as, with the increased inclusion of data it became impossible for us to explicitly state where changes had been made in line with your original comments that had often become surpassed by the inclusion of new data. In this latest revision we have included tracked changes and reference to lines amended, a practice that we would normally do as standard where revisions aren't extensive.

Line 37: They say "Critically, genetic assessment of sequences from mammalian species demonstrates no increased risk to human populations."

We have added a line to the abstract to state that mutations detected do not increase zoonotic risk over other observations from mammalian infection globally. (Line 38)

And again later in Line 329: "Critically, assessment of sequences derived from mammalian species did not indicate any adaptive mutations or increased risk to human populations. A single sequence from a single Southern elephant Seal sequence from Jason Harbour) contained the PB2 E627K mutation that is associated with adaptation to replication in mammalian species..." My issue is: how can you claim no increased risk to human populations when you find one of *THE* most important markers of mammalian host adaptation/pathogenicity, which is PB2 E627K present in a mammal?! Is it because you only find it in one sample? Isn't it also true that you recognize that you are vastly under-sampled? Your trees show it! Not to mention issues with your sample collection/testing. It doesn't escape me that working in Antarctica is incredibly difficult, and the logistics perhaps sometimes insurmountable. But the explanations provided to rebut the good critical feedback that was given during the first round of reviews are disappointing.

We thank the reviewer for these comments. The detection of E627K is a critical residue that has often been linked with the earliest adaptive response to replication in a non-avian host. However, whilst that fact is true, adaptation to mammals is generally considered to require further changes within PB2 as part of an accumulation of mutations. Importantly, we have sampled seals from the

beginning of the infection event on the islands as well as several weeks later and haven't seen a further accumulation in mutations in PB2. Further, and most importantly, we haven't observed any changes in HA associated with altered binding efficiency to sialic acid residues which are critical for increased zoonotic risk. Regardless we have altered the manuscript to further elaborate on this point. Lines 327 to 329.

For example, Line 258: "Information to date suggests that HPAIV infection in seals often leads to a neurological presentation with infrequent detection of viral material being detected through standard swab sampling activities. This may explain the initial lack of influenza vRNA detection in elephant seal swab samples taken in this study." Claiming "infrequent detection" in swabs is misleading. What is true is that the brains of infected animals have higher viral loads than other tissues, especially when they have encephalitis, but that does not mean you cannot pull this from swabs (or other tissues), so a more thoughtful and honest discussion of why you didn't or couldn't would be appreciated.

We agree with this point and have amended the text accordingly to state that the neurological presentation seen may affect shedding of infectious material/vRNA and hence detection through swab material (Line 263-264).

Another point: I asked you to include minimum coverage depth for your sequences, but you claim that this has "little relevance" since you are reporting consensus genomes. I completely disagree that coverage depth is of little relevance, even for consensus genomes. Yes, we often translate NGS data into consensus genomes, particularly in a situation like this one, where one is not looking at intra-host viral diversity, but a consensus genome with a coverage of 1x is completely different from a consensus genome with coverage of 10,000x. Coverage depth is a measure of certainty, of probability (in a sense) that the sequence you are reporting is correct, so for you to dismiss this as "not relevant" is not appropriate. Asking for coverage depth is the least I could ask you for. I could be asking you to deposit your raw data into public repositories for others to scrutinize, and yet, because of the gravity of the situation I am asking for simple information that you already have. Why do you not want to show it? Is your data not solid? Or is it that you don't trust it? Or do you not find it compelling enough? I would suspect that it is because there are parts of the genomes where your coverage depth is zero –if so, this should be discussed, not hidden. We all generate data where some parts of the genome have low or zero coverage depth. I went to GISAID to look for the sequences and they were hard to find (you also don't provide accession numbers), so along with the fact that you claim "no increased risk to humans" despite having E627K and being under-sampled worries me tremendously.

We have no problem with providing material as necessary and all sequences are ready to be released on GISAID. We have included the requested data as a supplementary table- Supplementary table S5. (Line 226-227)

Finally, line 340: "should the situation change..." I would argue the situation is already changing, only that you are catching it early and therefore you don't have all the evidence clearly displayed in front of you (just a few previews). And rather than seeing the situation for what it is (you have E627K!!!! how much more compelling can it get?!?!?!), you don't grasp its potential severity, and the

potentially critical finding of E627K in a mammal. Yes, it is true you only find it only in one sample thus far, but if it is in one sample already, it is a matter of time before it is found in others.

Thank you for this comment. We agree that the situation is dynamic and ever changing. However, the data included in the revised manuscript includes that up to the peak of the outbreak, a fact we can state now that the situation on the islands has thankfully receded. We have addressed the comments regarding zoonotic risk in the earlier comment. We will continue to assess further samples from a surveillance perspective over the coming months that may give further information about adaptation either to mammalian or different avian species. Critically, we have removed the phrasing 'Should this situation change'. (Line 337)

Reviewer #3 (Remarks to the Author):

I appreciate the opportunity to review the revised manuscript, 'Detection and spread of high pathogenicity avian influenza virus H5N1 in the Antarctic Region' submitted to Nature Communications. The authors have substantially reworked their original submission in response to three reviewer critiques. Additionally, the authors have expanded the scope of the manuscript to include numerous additional detections of highly pathogenic avian influenza (HPAI) viruses in the Antarctic Region. They have also bolstered analyses through the inclusion of additional genetic information on HPAI viruses. The manuscript is much improved and also substantially altered as compared to the original submission.

Thank you for this complementary comment.

From my perspective, the revised manuscript is conceptually sound and lacking any fatal flaws in reporting of information or interpretation. My assessment is that modest revisions would help to shore up several minor deficiencies within the narrative. I've appended (below) numerous comments towards this end. I thank the authors for their important contribution to Nature Communications and the broader scientific literature.

We thank the review for these kind words.

Lines 45-46: It is a gross over-simplification to state that GsGd HPAIV 'spread to other Asian countries'. Such viruses also spread through Europe, much of Africa, and into North America prior to 2021. Please revise this statement to make more congruent with the preceding and following statements.

We agree and have amended the statement and included a more suitable reference (Line 46)

Line 87: It is not explained how H11N2 viruses contrast with those of H4-H6 HA subtypes.

The H11 AIVs have been repeatably detected in the Antarctic region over several year periods, where these viruses show a high degree of genetic similarity to each other, indicating that they are being maintained in local bird populations. In contrast, H4-H6 subtypes sporadically detected in the Antarctic region show a high degree of genetic similarity to viruses detected in South America,

indicating that they have likely incurred into the Antarctic region from South America. (Lines 88-89)

Line 89: Perhaps spread of low pathogenic avian influenza viruses to the Antarctic Region from both the Americas and elsewhere provides evidence on HPAI virus introduction risk? See prior comment.

We have amended the text to incorporate the suggestion above (Line 88-89)

Line 115: I believe 'preyed', rather than predated, may be the more appropriate term here.

Amended as suggested. (Line 116)

Line 142: 6/17 mammalian carcasses are reported as HPAI virus positive in tables 1 and S2.

Amended as suggested. (Line 143)

Line 178-180: There is a grammatical issue in this statement.

We have amended this small grammatical error deleting the repeated 'have generated' statement- (Lines 179-181).

Line 188-189; I understand the 'full data set' to include only sequences generated in this study and not the representative sequences included in the maximum likelihood phylogenetic analyses as described throughout the preceding statements in the paragraph. As such, the statement explaining comparisons of nucleotide identity should probably be presented first with explanation of maximum likelihood phylogenetic analyses following thereafter.

We have reworded and reordered the text to take this factor into account. (Lines 182-191)

Line 192: The full tree (Figure 2A) with complete tip labels including strain names and/or accession numbers should be provided to readers for transparency and so the analysis can be replicated. Such a figure could be provided as a supplemental file.

We have generated this file as requested and placed it as a supplementary figure 2 and referenced it on Line 194 and Line 487.

Line 226: There is a tense issue with the verb 'has not' (e.g., should be 'had' not given the report of HPAI in the Antarctic Region in this product).

We have amended this error. (Line 230)

Line 262-264: I'm not understanding the logic that invasive sampling demonstrated that swab samples were positive. Are the authors trying to convey that testing of tissue samples collected through necropsy was often consistent with diagnostic results for swab samples generated from the same individuals?

Initial swab samples were negative for viral material yet on a later sampling, where we were able to get an APHA vet on the ground to undertake invasive sampling gave positivity from tissue samples. We have added a line to underscore the utility of tissues sampling. Line 268.

Lines 266-268, 289-295: I encourage the authors to focus discussion on information presented in

their report, not unreported/unpublished/unreviewed observations that cannot be verified by readers.

Lines 266-268: We have removed the text as suggested.

Lines 289-295: We have removed reference to the Leon manuscript as well as unpublished data by simply deleting this text

Line 304: The phrasing 'the original H5N1' is not appropriate nor correct. Please rephrase along the lines of, 'genotype 3.2 emerged following co-infection (of a presumably avian host) with a viral descendant of the first GsGd HPAI virus detected in North America during 2021 (purportedly introduced from Europe) with a North American lineage virus'.

We have amended the text as suggested. (Lines 299-302)

REVIEWERS' COMMENTS

Reviewer #1 (Remarks to the Author):

All of my comments have been addressed. Thank you. Great work!